# Multi-View Decision Processes:
# The Helper-AI Problem

**Christos Dimitrakakis**
Chalmers University of Technology & University of Lille
christos.dimitrakakis@gmail.com

**David C. Parkes**
Harvard University
parkes@eecs.harvard.edu

**Goran Radanovic**
Harvard University
gradanovic@g.harvard.edu

**Paul Tylkin**
Harvard University
ptylkin@g.harvard.edu

## Abstract

We consider a two-player sequential game in which agents have the same reward function but may disagree on the transition probabilities of an underlying Markovian model of the world. By committing to play a specific policy, the agent with the correct model can steer the behavior of the other agent, and seek to improve utility. We model this setting as a *multi-view decision process*, which we use to formally analyze the positive effect of steering policies. Furthermore, we develop an algorithm for computing the agents' achievable joint policy, and we experimentally show that it can lead to a large utility increase when the agents' models diverge.

## 1  Introduction.

In the past decade, we have been witnessing the fulfillment of Licklider's profound vision on AI [Licklider, 1960]:

> *Man-computer symbiosis is an expected development in cooperative interaction between men and electronic computers.*

Needless to say, such a collaboration, between humans and AIs, is natural in many real-world AI problems. As a motivating example, consider the case of autonomous vehicles, where a human driver can override the AI driver if needed. With advances in AI, the human will benefit most if she allows the AI agent to assume control and drive optimally. However, this might not be achievable—due to human behavioral biases, such as over-weighting the importance of rare events, the human might incorrectly override the AI. In the way, the misaligned models of the two drivers can lead to a decrease in utility. In general, this problem may occur whenever two agents disagree on their view of reality, even if they cooperate to achieve a common goal.

Formalizing this setting leads to a class of sequential multi-agent decision problems that extend stochastic games. While in a stochastic game there is an underlying transition kernel to which all agents (players) agree, the same is not necessarily true in the described scenario. Each agent may have a different transition model. We focus on a *leader-follower* setting in which the leader commits to a policy that the follower then best responds to, according to the follower's model. Mapped to our motivating example, this would mean that the AI driver is aware of human behavioral biases and takes them into account when deciding how to drive.

To incorporate both sequential and stochastic aspects, we model this as a *multi-view decision process*. Our multi-view decision process is based on an MDP model, with two, possibly different, transition kernels. One of the agents, hereafter denoted as $\mathscr{P}_1$, is assumed to have the correct transition kernel and is chosen to be the leader of the Stackelberg game—it commits to a policy that the second agent

($\mathscr{P}_2$) best-responds to according to its own model. The agents have the same reward function, and are in this sense *cooperative*. In an application setting, while the human ($\mathscr{P}_2$) may not be a planner, we motivate our set-up as modeling the endpoint of an adaptive process that leads $\mathscr{P}_2$ to adopt a best-response to the policy of $\mathscr{P}_1$.

Using the multi-view decision process, we analyze the effect of $\mathscr{P}_2$'s imperfect model on the achieved utility. We place an upper bound on the utility loss due to this, and also provide a lower bound on how much $\mathscr{P}_1$ gains by knowing $\mathscr{P}_2$'s model. One of our main analysis tools is the amount of *influence* an agent has, i.e. how much its actions affect the transition probabilities, both according to its own model, and according to the model of the other agent. We also develop an algorithm, extending backwards induction for simultaneous-move sequential games [c.f. Bošanský et al., 2016], to compute a pair of policies that constitute a subgame perfect equilibrium.

In our experiments, we introduce *intervention games* as a way to construct example scenarios. In an intervention game, an AI and a human share control of a process, and the human can intervene to override the AI's actions but suffers some cost in doing so. This allows us to derive a multi-view process from any single-agent MDP. We consider two domains: first, the intervention game variant of the *shelter-food game* introduced by Guo et al. [2013], as well as an autonomous driving problem that we introduce here. Our results show that the proposed approach provides a large increase in utility in each domain, thus overcoming the deficiencies of $\mathscr{P}_2$'s model, when the latter model is known to the AI.

## 1.1 Related work

*Environment design* [Zhang et al., 2009, Zhang and Parkes, 2008] is a related problem, where a first agent seeks to modulate the behavior of a second agent. However, the interaction between agents occurs through finding a good modification of the second agent's reward function: the AI observes a human performing a task, and uses inverse reinforcement learning [Ng et al., 2000] to estimate the human's reward function. Then it can assign extrinsic reward to different states in order to improve the human's policy. A similar problem in single-agent reinforcement learning is how to use internal rewards to improve the performance of a computationally-bounded, reinforcement learning agent [Sorg et al., 2010]. For example, even a myopic agent can maximize expected utility over a long time horizon if augmented with appropriately designed internal rewards. Our model differs from these prior works, in that the interaction between a 'helper agent' and a second agent is through taking actions in the same environment as the second agent.

In *cooperative inverse reinforcement learning* [Hadfield-Menell et al., 2016], an AI wants to cooperate with a human but does not initially understand the task. While their framework allows for simultaneous moves of the AI and the human, they only apply it to two-stage games, where the human demonstrates a policy in the first stage and the AI imitates in the second stage. They show that the human should take into account the AI's best response when providing demonstrations, and develop an algorithm for computing an appropriate demonstration policy. Our focus is on joint actions in a multi-period, uncertain environment, rather than teaching. The model of Amir et al. [2016] is also different, in that it considers the problem of how a teacher can optimally give advice to a sub-optimal learner, and is thus focused on communication and adaptation rather than interaction through actions. Finally, Elmalech et al. [2015] consider an advice-giving AI in single-shot games, where the human has an incorrect model. They experimentally find that when the AI heuristically models human expectations when giving advice, their performance is improved. We find that this also holds in our more general setting.

We cannot use standard methods for computing optimal strategies in stochastic games [Bošanský et al., 2015, Zinkevich et al., 2005], as the two agents have different models of the transitions between states. On the other extreme, a very general formalism to represent agent beliefs, such as that of Gal and Pfeffer [2008] is not well suited, because we have a Stackelberg setting and the problem of the follower is standard. Our approach is to extend backwards induction [c.f. Bošanský et al., 2016, Sec. 4] to the case of misaligned models in order to obtain a subgame perfect policy for the AI.

**Paper organization.**    Section 2 formalises the setting and its basic properties, and provides a lower bound on the improvement $\mathscr{P}_1$ obtains when $\mathscr{P}_2$'s model is known. Section 3 introduces a backwards induction algorithm, while Section 4 discusses the experimental results. We conclude with Section 5. Finally, Appendix A collects all the proofs, additional technical material and experimental details.

## 2   The Setting and Basic Properties

We consider two-agent sequential stochastic game, with two agents $\mathcal{P}_1, \mathcal{P}_2$, who disagree on the underlying model of the world, with the $i$-th agent's model being $\mu_i$, but share the same reward function. More formally,

**Definition 1** (Multi-view decision process (MVDP)). *A multi-view decision process $\mathcal{G} = \langle \mathcal{S}, \mathcal{A}, \sigma_1, \sigma_2, \mu_1, \mu_2, \rho, \gamma \rangle$ is a game between two agents, $\mathcal{P}_1$, $\mathcal{P}_2$, who share the same reward function. The game has a state space $\mathcal{S}$, with $S \triangleq |\mathcal{S}|$, action space $\mathcal{A} = \prod_i \mathcal{A}_i$, with $A \triangleq |\mathcal{A}|$, starting state distribution $\sigma$, transition kernel $\mu$, reward function[1] $\rho : \mathcal{S} \to [0, 1]$, and discount factor $\gamma \in [0, 1]$.*

*At time $t$, the agents observe the state $s_t$, take a joint action $\boldsymbol{a}_t = (a_{t,1}, a_{t,2})$ and receive reward $r_t = \rho(s_t)$. However, the two agents may have a different view of the game, with agent $i$ modelling the transition probabilities of the process as $\mu_i(s_{t+1} \mid s_t, \boldsymbol{a}_t)$ for the probability of the next state $s_{t+1}$ given the current state $s_t$ and joint action $\boldsymbol{a}_t$. Each agent's actions are drawn from a policy $\pi_i$, which may be an arbitrary behavioral policy, fixed at the start of the game. For a given policy pair $\boldsymbol{\pi} = (\pi_1, \pi_2)$, with $\pi_i \in \Pi_i$ and $\boldsymbol{\Pi} \triangleq \prod_i \Pi_i$, the respective payoff from the point of view of the $i$-th agent $u_i : \Pi \to \mathbb{R}$ is defined to be:*

$$u_i(\boldsymbol{\pi}) = \mathbb{E}_{\mu_i}^{\boldsymbol{\pi}}[U \mid s_1 \sim \sigma], \qquad U \triangleq \sum_{t=t}^{T} \gamma^{t-1} \rho(s_t). \tag{2.1}$$

For simplicity of presentation, we define *reward* $r_t = \rho(s_t)$ at time $t$, as a function of the state only, although an extension to state-action reward functions is trivial. The reward, as well, as well as the *utility $U$* (the discounted sum of rewards over time) are the same for both agents for a given sequence of states. However, the *payoff* for agent $i$ is their expected utility under the model $i$, and can be different for each agent.

Any two-player stochastic game can be cast into an MVDP:

**Lemma 1.** *Any two-player general-sum stochastic game (SG) can be reduced to a two-player MVDP in polynomial time and space.*

The proof of Lemma 1 is in Appendix A.

### 2.1   Stackelberg setting

We consider optimal policies from the point of view of $\mathcal{P}_1$, who is trying to assist a misguided $\mathcal{P}_2$. For simplicity, we restrict our attention to the Stackelberg setting, i.e. where $\mathcal{P}_1$ commits to a specific policy $\pi_1$ at the start of the game. This simplifies the problem for $\mathcal{P}_2$, who can play the optimal response according to the agent's model of the world. We begin by defining the (potentially unachievable) optimal joint policy, where both policies are chosen to maximise the same utility function:

**Definition 2** (Optimal joint policy). *A joint policy $\bar{\boldsymbol{\pi}}$ is optimal under $\sigma$ and $\mu_1$ iff $u_1(\bar{\boldsymbol{\pi}}) \geq u_1(\boldsymbol{\pi})$, $\forall \boldsymbol{\pi} \in \boldsymbol{\Pi}$. We furthermore use $\bar{u}_1 \triangleq u_1(\bar{\boldsymbol{\pi}})$ to refer to the value of the jointly optimal policy.*

This value may not be achievable, even though the two agents share a reward function, as the second agent's model does not agree with the first agent's, and so their expected utilities are different. To model this, we define the *Stackelberg utility* of policy $\pi_1$ for the first agent as:

$$u_1^{\text{St}}(\pi_1) \triangleq u_1(\pi_1, \pi_2^{\text{B}}(\pi_1)), \qquad \pi_2^{\text{B}}(\pi_1) = \arg\max_{\pi_2 \in \Pi_2} u_2(\pi_1, \pi_2), \tag{2.2}$$

i.e. the value of the policy when the second agent best responds to agent one's policy under the second agent's model.[2] The following defines the highest utility that $\mathcal{P}_1$ can achieve.

**Definition 3** (Optimal policy). *The optimal policy for $\mathscr{P}_1$, denoted by $\pi_1^*$, is the one maximizing the Stackelberg utility, i.e. $u_1^{St}(\pi_1^*) \geq u_1^{St}(\pi_1)$, $\pi_1 \in \Pi_1$, and we use $u_1^* \triangleq u^{St}(\pi_1^*)$ to refer to the value of this optimal policy.*

In the remainder of the technical discussion, we will characterize $\mathscr{P}_1$ policies in terms of how much worse they are than the jointly optimal policy, as well as how much better they can be than the policy that blithely assumes that $\mathscr{P}_2$ shares the same model.

We start with some observations about the nature of the game when one agent fixes its policy, and we argue how the difference between the models of the two agents affects the utility functions. We then combine this with a definition of *influence* to obtain bounds on the loss due to the difference in the models.

When agent $i$ fixes a Markov policy $\pi_i$, the game is an MDP for agent $j$. However, if agent $i$'s policy is not Markovian the resulting game is not an MDP on the original state space. We show that if $\mathscr{P}_1$ acts as if $\mathscr{P}_2$ has the correct transition kernel, then the resulting joint policy has value bounded by the $L_1$ norm between the true kernel and agent 2's actual kernel. We begin by establishing a simple inequality to show that knowledge of the model $\mu_2$ is beneficial for $\mathscr{P}_1$.

**Lemma 2.** *For any MVDP, the utility of the jointly optimal policy is greater than that of the (achievable) optimal policy, which is in turn greater than that of the policy that assumes that $\mu_2 = \mu_1$.*

$$u_1(\bar{\boldsymbol{\pi}}) \geq u_1^{St}(\pi_1^*) \geq u_1^{St}(\bar{\pi}_1) \tag{2.3}$$

*Proof.* The first inequality follows from the definition of the jointly optimal policy and $u_1^{St}$. For the second inequality, note that the middle term is a maximizer for the right-hand side. □

Consequently, $\mathscr{P}_1$ must be able to do (weakly) better if it knows $\mu_2$ compared to if it just assumes that $\mu_2 = \mu_1$. However, this does not tell us how much (if any) improvement we can obtain. Our idea is to see what policy $\pi_1$ we would need to play in order to make $\mathscr{P}_2$ play $\bar{\pi}_2$, and measure the distance of this policy from $\bar{\pi}_1$. To obtain a useful bound, we need to have a measure on how much $\mathscr{P}_1$ must deviate from $\bar{\pi}_1$ in order for $\mathscr{P}_2$ to play $\bar{\pi}_2$. For this, we define the notion of *influence*. This will capture the amount by which a agent $i$ can affect the game in the eyes of agent $j$. In particular, it is the maximal amount by which an agent $i$ can affect the transition distribution of agent $j$ by changing $i$'s action at each state $s$:

**Definition 4** (Influence). *The influence of agent $i$ on the transition distribution of model $\mu_j$ is defined as the vector:*

$$\mathcal{I}_{i,j}(s) \triangleq \max_{a_{t,-i}} \max_{a_{t,i} a'_{t,i}} \|\mu_j(\cdot \mid s_t = s, a_{t,i}, a_{t,-i}) - \mu_j(\cdot \mid s_t = s, a'_{t,i}, a_{t,-i})\|_1, \tag{2.4}$$

*where the norm is over the difference in next-state distributions $s_{t+1}$ for the two models.*

Thus, $\mathcal{I}_{1,1}$ describes the *actual* influence of $\mathscr{P}_1$ on the transition probabilities, while $\mathcal{I}_{1,2}$ describes the *perceived* influence of $\mathscr{P}_1$ by $\mathscr{P}_2$. We will use influence to define an $\mu$-dependent distance between policies, capturing the effect of an altered policy on the model:

**Definition 5** (Policy distance). *The distance between policies $\pi_i, \pi_i'$ under model $\mu_j$ is:*

$$\|\pi_i - \pi_i'\|_{\mu_j} \triangleq \max_{s \in \mathcal{S}} \|\pi_i(\cdot \mid s) - \pi_i'(\cdot \mid s)\|_1 \mathcal{I}_{i,j}(s). \tag{2.5}$$

These two definitions result in the following Lipschitz condition on the utility function, whose proof can be found in Appendix A.

**Lemma 3.** *For any fixed $\pi_2$, and any $\pi_1, \pi_1'$: $u_i(\pi_1, \pi_2) \leq u_i(\pi_1', \pi_2) + \|\pi_1 - \pi_1'\|_{\mu_i} \frac{\gamma}{(1-\gamma)^2}$, with a symmetric result holding for any fixed policy $\pi_1$, and any pair $\pi_2, \pi_2'$.*

Lemma 3 bounds the change in utility due to a change in policy by $\mathscr{P}_1$ with respect to $i$'s payoff. As shall be seen in the next section, it allows us to analyze how close the utility we can achieve comes to that of the jointly optimal policy, and how much can be gained by not naively assuming that the model of $\mathscr{P}_2$ is the same.

## 2.2 Optimality

In this section, we illuminate the relationship between different types of policies. First, we show that if $\mathscr{P}_1$ simply assumes $\mu_2 = \mu_1$, it only suffers a bounded loss relative to the jointly optimal policy. Subsequently, we prove that knowing $\mu_2$ allows $\mathscr{P}_1$ to find an improved policy.

**Lemma 4.** *Consider the optimal policy $\bar{\pi}_1$ for the modified game $\widehat{\mathcal{G}} = \langle \mathcal{S}, \mathcal{A}, \sigma_1, \sigma_1, \mu_1, \mu_1, \rho, \gamma \rangle$ where $\mathscr{P}_2$'s model is correct. Then $\bar{\pi}_1$ is Markov and achieves utility $\bar{u}$ in $\widehat{G}$, while its utility in $\mathcal{G}$ is:*

$$u_1^{St}(\bar{\pi}_1) \geq \bar{u} - \frac{2\|\mu_1 - \mu_2\|_1}{(1-\gamma)^2}, \qquad \|\mu_1 - \mu_2\|_1 \triangleq \max_{s_t, \boldsymbol{a}_t} \|\mu_1(s_{t+1} \mid s_t, \boldsymbol{a}_t) - \mu_2(s_{t+1} \mid s_t, \boldsymbol{a}_t)\|_1.$$

As this bound depends on the maximum between all state action pairs, we refine it in terms of the influence of each agent's actions. This also allows us to measure the loss in terms of the difference in $\mathscr{P}_2$'s actual and desired response, rather than the difference between the two models, which can be much larger.

**Corollary 1.** *If $\mathscr{P}_2$'s best response to $\bar{\pi}_1$ is $\pi_2^B(\bar{\pi}_1) \neq \bar{\pi}_2$, then our loss relative to the jointly optimal policy is bounded by $u_1(\bar{\pi}_1, \bar{\pi}_2) - u_1(\bar{\pi}_1, \pi_2^B(\bar{\pi}_1)) \leq \left\|\pi_2^B(\bar{\pi}_1) - \bar{\pi}_2\right\|_{\mu_1} \frac{\gamma}{(1-\gamma)^2}$.*

*Proof.* This follows from Lemma 3 by fixing $\bar{\pi}_1$ for the policy pairs $\pi_2^B(\bar{\pi}_1), \bar{\pi}_2$ under $\mu_1$. $\qquad\square$

While the previous corollary gave us an upper bound on the loss we incur if we ignore the beliefs of $\mathscr{P}_2$, we can bound the loss of the optimal Stackleberg policy in the same way:

**Corollary 2.** *The difference between the optimal utility $u_1(\bar{\pi}_1, \bar{\pi}_2)$ and the optimal Stackleberg utility $u_1^{St}(\pi_1^*)$ is bounded by $u_1(\bar{\pi}_1, \bar{\pi}_2) - u_1^{St}(\pi_1^*) \leq \left\|\pi_2^B(\bar{\pi}_1) - \bar{\pi}_2\right\|_{\mu_1} \frac{\gamma}{(1-\gamma)^2}$.*

*Proof.* The result follows directly from Corollary 1 and Lemma 2. $\qquad\square$

This bound is not very informative by itself, as it does not suggest an advantage for the optimal Stackelberg policy. Instead, we can use Lemma 3 to lower bound the increase in utility obtained relative to just playing the optimistic policy $\bar{\pi}_1$. We start by observing that when $\mathscr{P}_2$ responds with some $\hat{\pi}_2$ to $\bar{\pi}_1$, $\mathscr{P}_1$ could improve upon this by playing $\hat{\pi}_1 = \pi_1^B(\hat{\pi}_2)$, the best response of to $\hat{\pi}_2$, if $\mathscr{P}_1$ could somehow force $\mathscr{P}_2$ to stick to $\hat{\pi}_2$. We can define

$$\Delta \triangleq u_1(\hat{\pi}_1, \hat{\pi}_2) - u_1(\bar{\pi}_1, \hat{\pi}_2), \tag{2.6}$$

to be the *potential advantage* from switching to $\hat{\pi}_1$. Theorem 1 characterizes how close to this advantage $\mathscr{P}_1$ can get by playing a stochastic policy $\pi_1^\alpha(a \mid s) \triangleq \alpha\bar{\pi}_1(a \mid s) + (1-\alpha)\hat{\pi}_1(a \mid s)$, while ensuring that $\mathscr{P}_2$ sticks to $\hat{\pi}_2$.

**Theorem 1** (A sufficient condition for an advantage over the naive policy). *Let $\hat{\pi}_2 = \pi_2^B(\bar{\pi}_1)$ be the response of $\mathscr{P}_2$ to the optimistic policy $\bar{\pi}_1$ and assume $\Delta > 0$. Then we can obtain an advantage of at least:*

$$\Delta - \frac{\gamma \left\|\bar{\pi}_1 - \hat{\pi}_1\right\|_{\mu_1}}{(1-\gamma)^2} + \frac{\delta}{2} \frac{\left\|\bar{\pi}_1 - \hat{\pi}_1\right\|_{\mu_1}}{\left\|\bar{\pi}_1 - \hat{\pi}_1\right\|_{\mu_2}} \tag{2.7}$$

*where $\delta \triangleq u_2(\bar{\pi}_1, \hat{\pi}_2) - \max_{\pi_2 \neq \hat{\pi}_2} u_2(\bar{\pi}_1, \pi_2)$ is the gap between $\hat{\pi}_2$ and all other deterministic policies of $\mathscr{P}_2$ when $\mathscr{P}_1$ plays $\bar{\pi}_1$.*

We have shown that knowledge of $\mu_2$ allows $\mathscr{P}_1$ to obtain improved policies compared to simply assuming $\mu_2 = \mu_1$, and that this improvement depends on both the real and perceived effects of a change in $\mathscr{P}_1$'s policy. In the next section we develop an efficient dynamic programming algorithm for finding a good policy for $\mathscr{P}_1$.

# 3 Algorithms for the Stackelberg Setting

In the Stackelberg setting, we assume that $\mathscr{P}_1$ commits to a policy $\pi_1$, and this policy is observed by $\mathscr{P}_2$. Because of this, it is sufficient for $\mathscr{P}_2$ to use a Markov policy, and this can be calculated in polynomial time in the number of states and actions.

However, there is a polynomial reduction from stochastic games to MVDPs (Lemma 1), and since Letchford et al. [2012] show that computing optimal commitment strategies is NP-hard, then the planning problem for MVDPs is also NP-hard. Another difficulty that occurs is that dominating policies in the MDP sense may not exist in MVDPs.

**Definition 6** (Dominating policies)**.** A dominating policy $\pi$ satisfies $V^\pi(s) \geq V^{\pi'}(s), \forall s \in \mathcal{S}$, where $V^\pi(s) = \mathbb{E}^\pi(u \mid s_0 = s)$.

Dominating policies have the nice property that they are also optimal for any starting distribution $\sigma$. However, dominating, stationary Markov polices need not exist in our setting.

**Theorem 2.** *A dominating, stationary Markov policy may not exist in a given MVDP.*

The proof of this theorem is given by a counterexample in Appendix A, where the optimal policy depends on the history of previously visited states.

In the trivial case when $\mu_1 = \mu_2$, the problem can be reduced to a Markov decision process, which can be solved in $O(S^2 A)$ [Mansour and Singh, 1999, Littman et al., 1995]. Generally, however, the commitment by $\mathscr{P}_1$ creates new dependencies that render the problem inherently non-Markovian with respect to the state $s_t$ and thus harder to solve. In particular, even though the dynamics of the environment are Markovian with respect to the state $s_t$, the MVDP only becomes Markov in the Stackelberg setting with respect to the hyper-state $\eta_t = (s_t, \pi_{t:T,1})$ where $\pi_{t:T,1}$ is the commitment by $\mathscr{P}_1$ for steps $t, \ldots, T$. To see that the game is non-Markovian, we only need to consider a single transition from $s_t$ to $s_{t+1}$. $\mathscr{P}_2$'s action depends not only on the action $a_{t,1}$ of $\mathscr{P}_1$, but also on the expected utility the agent will obtain in the future, which in turn depends on $\pi_{t:T,1}$. Consequently, state $s_t$ is not a sufficient statistic for the Stackelberg game.

## 3.1 Backwards Induction

These difficulties aside, we now describe a backwards induction algorithm for approximately solving MVDPs. The algorithm can be seen as a generalization of the backwards induction algorithm for simultaneous-move stochastic games [c.f. Bošanský et al., 2016] to the case of disagreement on the transition distribution.

In our setting, at stage $t$ of the interaction, $\mathscr{P}_2$ has observed the current state $s_t$ and also knows the commitment of $\mathscr{P}_1$ for all future periods. $\mathscr{P}_2$ now chooses the action

$$a_{t,2}^*(\pi_1) \in \arg\max_{a_{t,2}} \rho(s_t) + \gamma \sum_{a_{t,1}} \pi_1(a_{t,1} \mid s_t) \sum_{s_{t+1}} \mu_2(s_{t+1}|s_t, a_{t,1}, a_{t,2}) \cdot V_{2,t+1}(s_{t+1}). \quad (3.1)$$

Thus, for every state, there is a well-defined continuation for $\mathscr{P}_2$. Now, $\mathscr{P}_1$ needs to choose an action. This can be done easily, since we know $\mathscr{P}_2$'s continuation, and so we can define a value for each state-action-action triplet for either agent:

$$Q_{i,t}(s_t, a_{t,1}, a_{t,2}) = \rho(s) + \gamma \sum_{s_{t+1}} \mu_i(s_{t+1}|s_t, a_{t,1}, a_{t,2}) \cdot V_{i,t+1}(s_{t+1}).$$

As the agents act simultaneously, the policy of $\mathscr{P}_1$ needs to be stochastic. The local optimization problem can be formed as a set of linear programs (LPs), one for each action $a_2 \in \mathcal{A}_2$:

$$\max_{\pi_1} \sum_{a_1} \pi_1(a_1|s) \cdot Q_{t,1}(s, a_1, a_2)$$

$$\text{s.t. } \forall \hat{a}_2 : \sum_{a_1} \pi_1(a_1|s) \cdot Q_{t,2}(s, a_1, a_2) \geq \sum_{a_1} \pi(a_1) \cdot Q_{t,2}(s, a_1, \hat{a}_2),$$

$$\forall \hat{a}_1 : 0 \leq \pi_1(\hat{a}_1|s) \leq 1, \text{ and } \sum_{a_1} \pi_1(a_1|s) = 1.$$

Each LP results in the best possible policy at time $t$, such that we force $\mathscr{P}_2$ to play $a_2$. From these, we select the best one. At the end, the algorithm, given the transitions $(\mu_1, \mu_2)$, and the time horizon $T$, returns an approximately optimal joint policy, $(\pi_1^*, \pi_2^*)$ for the MVDP. The complete pseudocode is given in Appendix C, algorithm 1.

As this solves a finite horizon problem, the policy is inherently non-stationary. In addition, because there is no guarantee that there is a dominating policy, we may never obtain a stationary policy (see below). However, we can extract a stationary policy from the policies played at individual time steps $t$, and select the one with the highest expected utility. We can also obtain a version of the algorithm that attains a deterministic policy, by replacing the linear program with a maximization over $\mathscr{P}_1$'s actions.

**Optimality.** The policies obtained using this algorithm are subgame perfect, up to the time horizon adopted for backward induction; i.e. the continuation policies are optimal (considering the possibly incorrect transition kernel of $\mathscr{P}_2$) off the equilibrium path. As a dominating Markov policy may not exist, the algorithm may not converge to a stationary policy in the infinite horizon discounted setting, similarly to the cyclic equilibria examined by Zinkevich et al. [2005]. This is because the commitment of $\mathscr{P}_1$ affects the current action of $\mathscr{P}_2$, and so the effective transition matrix for $\mathscr{P}_1$. More precisely, the transition actually depends on the future joint policy $\pi^{n+1:T}$, because this determines the value $Q_{2,t}$ and so the policy of $\mathscr{P}_2$. Thus, the Bellman optimality condition does not hold, as the optimal continuation may depend on previous decisions.

## 4 Experiments

We focus on a natural subclass of multi-view decision processes, which we call *intervention games*. Therein, a human and an AI have joint control of a system, and the human can override the AI's actions at a cost. As an example, consider semi-autonomous driving, where the human always has an option to override the AI's decisions. The cost represents the additional effort of human intervention; if there was no cost, the human may always prefer to assume manual control and ignore the AI.

**Definition 7** (*c*-intervention game)**.** A MVDP is a $c$-intervention game if all of $\mathscr{P}_2$'s actions override those of $\mathscr{P}_1$, apart from the null action $a^0 \in \mathcal{A}_2$, which has no effect.

$$\mu_1(s_{t+1} \mid s_t, a_{t,1}, a_{t,2}) = \mu_1(s_{t+1} \mid s_t, a'_{t,1}, a_{t,2}) \qquad \forall a_{t,1}, a'_{t,1} \in \mathcal{A}, a_{t,2} \neq a^0. \qquad (4.1)$$

In addition, the agents subtract a cost $c(s) > 0$ from the reward $r_t = \rho(s_t)$ whenever $\mathscr{P}_2$ takes an action other than $a^0$.

Any MDP with action space $\mathcal{A}'$ and reward function $\rho' \colon \mathcal{S} \to [0, 1]$ can be converted into a $c$-intervention game, and modeled as an MVDP, with action space $\mathcal{A} = \mathcal{A}_1 \times \mathcal{A}_2$, where $\mathcal{A}_1 = \mathcal{A}'$, $\mathcal{A}_2 = \mathcal{A}_1 \cup \{a^0\}, a_1 \in \mathcal{A}_1, a_2 \in \mathcal{A}_2, a = (a_1, a_2) \in \mathcal{A}$,

$$r_{\text{MIN}} = \min_{s' \in S, \, a'_2 \in \mathcal{A}_2} \rho'(s') - c(s'), \qquad (4.2)$$

$$r_{\text{MAX}} = \max_{s' \in S, \, a'_2 \in \mathcal{A}_2} \rho'(s') \qquad (4.3)$$

and reward function[3] $\rho \colon \mathcal{S} \times \mathcal{A} \to [0, 1]$, with

$$\rho(s, a) = \frac{\rho'(s) - c(s) \, \mathbb{I}\{a_2 \neq a^0\} - r_{\text{MIN}}}{r_{\text{MAX}} - r_{\text{MIN}}}. \qquad (4.4)$$

The reward function in the MVDP is defined so that it also has the range $[0, 1]$.

**Algorithms and scenarios.** We consider the main scenario, as well as three variant scenarios, with different assumptions about the AI's model. For the main scenario, the human has an incorrect model of the world, which the AI knows. For this, we consider three types of AI policies:

PURE: The AI only uses deterministic Markov policies.

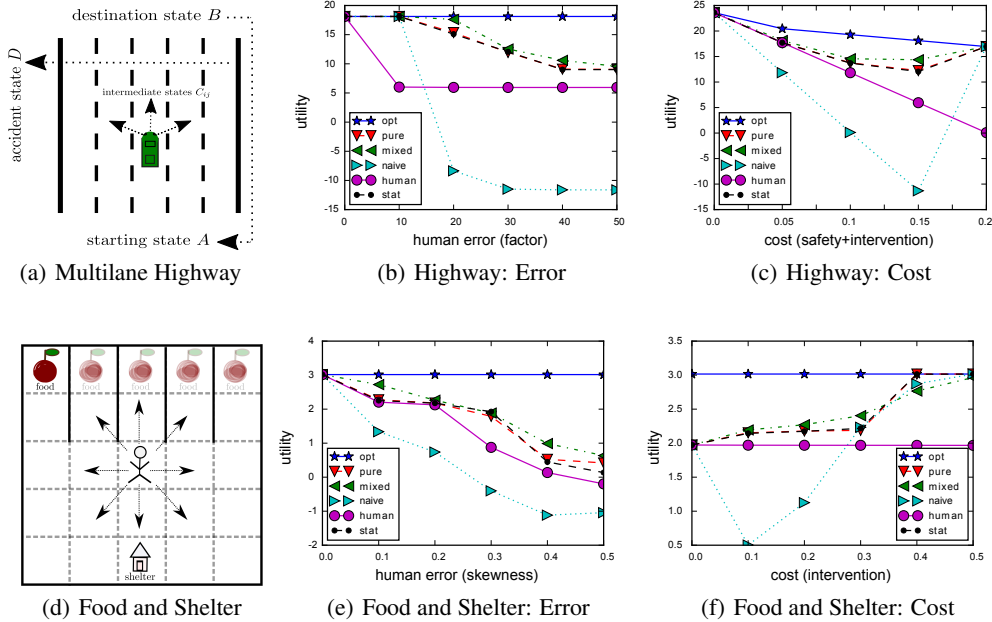

| (a) Multilane Highway | (b) Highway: Error | (c) Highway: Cost |
|---|---|---|
| (d) Food and Shelter | (e) Food and Shelter: Error | (f) Food and Shelter: Cost |

Figure 1: Illustrations and experimental results for the 'multilane highway' and 'food and shelter' domains. Plots (b,e) show the effect of varying the error in the human's transition kernel with fixed intervention cost. Plots (c,f) show the effect of varying the intervention cost for a fixed error in the human's transition kernel.

MIXED: The AI may use stochastic Markov policies.

STAT: As above, but use the best instantaneous deterministic policy of the first 25 time-steps found in PURE as a stationary Markov policy (running for the same time horizon as PURE).

We also have three variant scenarios of AI and human behaviour.

OPT: Both the AI and human have the correct model of the world.

NAIVE: The AI assumes that the human's model is correct.

HUMAN: Both agents use the incorrect human model to take actions. It is equivalent to the human having full control without any intervention cost.

In all of these, the AI uses a MIXED policy. We consider two simulated problem domains in which to evaluate our methods. The first is a *multilane highway scenario*, where the human and AI have shared control of a car, and the second is a *food and shelter domain* where they must collect food and maintain a shelter. In all cases, we use a finite time horizon of 100 steps and a discount factor of $\gamma = 0.95$.

**Multilane Highway.** In this domain, a car is under joint control of an AI agent and a human, with the human able to override the AI's actions at any time. There are multiple lanes in a highway, with varying levels of risk and speed (faster lanes are more risky). Within each lane, there is some probability of having an accident. However, the human overestimates this probability, and so wants to travel in a slower lane than is optimal. We denote a starting state by $A$, a destination state by $B$, and, for lane $i$, intermediate states $C_{i1}, ..., C_{iJ}$, where $J$ is the number of intermediate states in a lane, and an accident state $D$. See Figure 1(a) for an illustration of the domain, and for the simulation results. In the plots, the error parameter represents a factor by which the human is wrong in assessing the accident probability (assumed to be small), while the cost parameter determines both the cost of safety (slow driving) of different lanes as well as the cost of human intervening on these lanes. The latter is because our experimental model couples the cost of intervention with the safety cost. The rewards range from $-10$ to $10$. More details are provided in the Appendix (Section B).

**Food and Shelter Domain.** The food and shelter domain [Guo et al., 2013] involves an agent simultaneously trying to find randomly placed food (in one of the top five locations) while maintaining a shelter. With positive probability at each time step, the shelter can collapse if it is not maintained. There is a negative reward for the shelter collapsing and positive reward for finding food (food reappears whenever it is found). In order to exercise the abilities of our modeling, we make the original setting more complex by increasing the size of the grid to $5 \times 5$ and allowing diagonal moves. For our MVDP setting, we give the AI the correct model but assume the human overestimates the probabilities. Furthermore, the human believes that diagonal movements are more prone to error. See Figure 1(d) for an illustration of the domain, and for the simulation results. In the plots, the error parameter determines how skewed the human's belief about the error is towards the uniform distribution, while the cost parameter determines the cost of intervention. The rewards range from $-1$ to $1$. More details are provided in the Appendix (Section B).

**Results.** In the simulations, when we change the error parameter, we keep the cost parameter constant (0.15 for the multilane highway domain and 0.1 for the food and shelter domain), and vice versa, when we change the cost, we keep the error constant (25 for the multilane highway domain and 0.25 for the food and shelter domain). Overall, the results show that PURE, MIXED and STAT perform considerably better than NAIVE and HUMAN. Furthermore, for low costs, HUMAN is better than NAIVE. The reason is that in NAIVE the human agent overrides the AI, which is more costly than having the AI perform the same policy (as it happens to be for HUMAN). Therefore, simply assuming that the human has the correct model does not only lead to a larger error than knowing the human's model, but it can also be worse than simply adopting the human's erroneous model when making decisions.

As the cost of intervention increases, the utilities become closer to the jointly optimal one (OPT scenario), with the exception of the utility for scenario HUMAN. This is not surprising since the intervention cost has an important tempering effect—the human is less likely to take over the control if interventions are costly. When the human error is small, the utility approaches that of the jointly optimal policy. Clearly, the increasing error leads to larger deviations from the the optimal utility.

Out of the three algorithms (PURE, MIXED and STAT), MIXED obtains a slightly better performance and shows the additional benefit from allowing for stochastic polices. PURE and STAT have quite similar performance, which indicates that in most of the cases the backwards induction algorithm converges to a stationary policy.

# 5   Conclusion

We have introduced the framework of multi-view decision processes to model value-alignment problems in human-AI collaboration. In this problem, an AI and a human act in the same environment, and share the same reward function, but the human may have an incorrect world model. We analyze the effect of knowledge of the human's world model on the policy selected by the AI.

More precisely, we developed a dynamic programming algorithm, and gave simulation results to demonstrate that an AI with this algorithm can adopt a useful policy in simple environments and even when the human adopts an incorrect model. This is important for modern applications involving the close cooperation between humans and AI such as home robots or automated vehicles, where the human can choose to intervene but may do so erroneously. Although backwards induction is efficient for discrete state and action spaces, it cannot usefully be applied to the continuous case. We would like to develop stochastic gradient algorithms for this case. More generally, we see a number of immediate extensions to MVDP: estimating the human's world model, studying a setting in which human is learning to respond to the actions of the AI, and moving away from Stackelberg to the case of no commitment.

**Acknowledgements.** The research has received funding from: the People Programme (Marie Curie Actions) of the European Union's Seventh Framework Programme (FP7/2007-2013) under REA grant agreement 608743, the Swedish national science foundation (VR), the Future of Life Institute, the SEAS TomKat fund, and a SNSF Early Postdoc Mobility fellowship.

## Footnotes

[1]For simplicity we consider state-dependent rewards bounded in $[0, 1]$. Our results are easily generalizable to $\rho : \mathcal{S} \times \mathcal{A} \to [0, 1]$, through scaling by a factor of $B$ and shifting by a factor of $b$m for any reward function in $[b, b + B]$.

[2]If there is no unique best response, we define the utility in terms of the worst-case, best response.

[3]Note that although our original definition used a state-only reward function, we are using a state-action reward function.

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
