[Supplementary Material]

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

# A  Collected Proofs

*Proof of Lemma 1.* Let us start with an SG with transition kernel $\mu_{SG}$ and reward functions $\left\{ \rho_{SG}^1, \rho_{SG}^2 \right\}$ on some state space $\mathcal{S}_{SG}$ so that the reward of $\mathscr{P}_i$ at some state $s$ is $\rho_{SG}^i(s)$. W.l.o.g., suppose that the states of $\mathcal{S}_{SG}$ are enumerated as $0, ..., |\mathcal{S}_{SG}| - 1$. Now, let us define a new state space $\mathcal{S}_{MVDP}$ whose cardinality is $|\mathcal{S}_{MVDP}| = 2|\mathcal{S}_{SG}|$, and let the states of $\mathcal{S}_{MVDP}$ be enumerated as $0, ..., |\mathcal{S}_{MVDP}| - 1$. We define a new MVDP on state space $\mathcal{S}_{MVDP}$, with the same action space as SG, but with the set of transition kernels $\left\{ \mu_{MVDP}^1, \mu_{MVDP}^2 \right\}$ defined as:

$$\mu_{MVDP}^1(k_1|a_1, a_2, k_2) = \begin{cases} \mu_{SG}(k_1/2|a_1, a_2, \lfloor k_2/2 \rfloor) & \text{if } k_1 \mod 2 = 0 \\ 0 & \text{if } k_1 \mod 2 = 1 \end{cases},$$

$$\mu_{MVDP}^2(k_1|a_1, a_2, k_2) = \begin{cases} 0 & \text{if } k_1 \mod 2 = 0 \\ \mu_{SG}((k_1 - 1)/2|a_1, a_2, \lfloor k_2/2 \rfloor) & \text{if } k_1 \mod 2 = 1 \end{cases},$$

a reward function $\rho_{MVDP}$ defined as:

$$\rho_{MVDP}(k) = \begin{cases} \rho_{SG}^1(k/2) & \text{if } k \mod 2 = 0 \\ \rho_{SG}^2((k - 1)/2) & \text{if } k \mod 2 = 1 \end{cases},$$

and starting state distributions $\left\{ \sigma_{MVDP}^1, \sigma_{MVDP}^2 \right\}$ defined as:

$$\sigma_{MVDP}^1(k_1) = \begin{cases} \sigma_{SG}(k_1/2) & \text{if } k_1 \mod 2 = 0 \\ 0 & \text{if } k_1 \mod 2 = 1 \end{cases},$$

$$\sigma_{MVDP}^2(k_1) = \begin{cases} 0 & \text{if } k_1 \mod 2 = 0 \\ \sigma_{SG}((k_1 - 1)/2) & \text{if } k_1 \mod 2 = 1 \end{cases}.$$

In other words, states in the original SG are mirrored into two different hyperstates: one for $\mathscr{P}_1$ and one for $\mathscr{P}_2$, and the reward of each state is equal to the reward of the corresponding agent in the original state, while each agent believes that they can only transition to their own hyper-states, with transition probabilities remaining identical to those in SG. This implies that for any pair of policies $(\pi_1, \pi_2)$, the resulting expected utilities $u_1(\pi_1, \pi_2)$ and $u_2(\pi_1, \pi_2)$ will be the same for SG and MVDP. Therefore, the presented transformation defines a polynomial time and space reduction from an SG to an MVDP. $\square$

To prove Lemma 3 we need the following remark that relates the distance between policy to the MDP distance. All the related exposition is with respect to stationary policies. This is sufficient, since when the one agent has a stationary policy, the other's optimal response is always stationary for the undiscounted infinite horizon setting.

**Lemma 5.** *For any two policies* $\pi_1, \pi_1'$ *of* $\mathscr{P}_1$, *the resulting transition probability matrices* $P_\mu^{\pi_1, \pi_2}(s'|s) \triangleq \sum_{a_1, a_2} \mu(s'|s, a_1, a_2) \cdot \pi_1(a_1|s) \cdot \pi_2(a_2|s)$:

$$\|\pi_1 - \pi_1'\|_\mu \geq \|P_\mu^{\pi_1, \pi_2} - P_\mu^{\pi_1', \pi_2}\|_1 \qquad \forall \pi_2. \tag{A.1}$$

*Proof.* In the following we use $\boldsymbol{\pi} = (\pi_1, \pi_2)$ and $\boldsymbol{\pi}' = (\pi_1', \pi_2)$ to denote the two different joint policies that arise. We also compactly denote the two resulting transition matrices by $p \triangleq P_i^{\pi_1, \pi_2}$ and $p' \triangleq P_i^{\pi_1', \pi_2}$. The proof follows by elementary manipulations and norm inequalities:

$$\|p - p'\|_1 = \max_s \|p(\cdot|s) - p'(\cdot|s)\|_1$$

$$= \max_s \|\sum_{\boldsymbol{a}} \mu_i(\cdot|s, \boldsymbol{a})\boldsymbol{\pi}(\boldsymbol{a}|s) - \mu_i(\cdot|s, \boldsymbol{a})\boldsymbol{\pi}'(\boldsymbol{a}|s)\|_1$$

$$= \max_s \|\sum_{\boldsymbol{a}} \mu_i(\cdot|s, \boldsymbol{a})[\boldsymbol{\pi}(\boldsymbol{a}|s) - \boldsymbol{\pi}'(\boldsymbol{a}|s)]\|_1$$

$$= \max_s \|\sum_{a_1 \neq a_1', a_2|\boldsymbol{a} \triangleq (a_1, a_2)} \mu_i(\cdot|s, \boldsymbol{a})[\boldsymbol{\pi}(\boldsymbol{a}|s) - \boldsymbol{\pi}'(\boldsymbol{a}|s)] + \sum_{a_2|\boldsymbol{a}' \triangleq (a_1', a_2)} \mu_i(\cdot|s, \boldsymbol{a}')[\boldsymbol{\pi}(\boldsymbol{a}'|s) - \boldsymbol{\pi}'(\boldsymbol{a}'|s)]\|_1$$

$$= \max_s \|\sum_{a_1 \neq a_1', a_2} \mu_i(\cdot|s, \boldsymbol{a})[\boldsymbol{\pi}(\boldsymbol{a}|s) - \boldsymbol{\pi}'(\boldsymbol{a}|s)]$$

$$+ \sum_{a_2} \mu_i(\cdot|s, \boldsymbol{a}')[\pi_2(a_2|s) \cdot (1 - \sum_{a_1 \neq a_1'} \pi_1(a_1|s)) - \pi_2(a_2|s) \cdot (1 - \sum_{a_1 \neq a_1'} \pi_1'(a_1|s))]\|_1$$

$$= \max_s \|\sum_{a_1 \neq a_1', a_2} \mu_i(\cdot|s, \boldsymbol{a})[\boldsymbol{\pi}(\boldsymbol{a}|s) - \boldsymbol{\pi}'(\boldsymbol{a}|s)] - \sum_{a_2} \mu_i(\cdot|s, \boldsymbol{a}')[\sum_{a_1 \neq a_1'} \boldsymbol{\pi}(\boldsymbol{a}|s) - \sum_{a_1 \neq a_1'} \boldsymbol{\pi}'(\boldsymbol{a}|s)]\|_1$$

$$= \max_s \|\sum_{a_1 \neq a_1', a_2} [\mu_i(\cdot|s, \boldsymbol{a}) - \mu_i(\cdot|s, \boldsymbol{a}')][\boldsymbol{\pi}(\boldsymbol{a}|s) - \boldsymbol{\pi}'(\boldsymbol{a}|s)]\|_1$$

$$\leq \max_s \sum_{a_1 \neq a_1', a_2} \|[\mu_i(\cdot|s, \boldsymbol{a}) - \mu_i(\cdot|s, \boldsymbol{a}')][\boldsymbol{\pi}(\boldsymbol{a}|s) - \boldsymbol{\pi}'(\boldsymbol{a}|s)]\|_1$$

$$= \max_s \sum_{a_1 \neq a_1', a_2} \|\mu_i(\cdot|s, \boldsymbol{a}) - \mu_i(\cdot|s, \boldsymbol{a}')\|_1 \|\boldsymbol{\pi}(\boldsymbol{a}|s) - \boldsymbol{\pi}'(\boldsymbol{a}|s)\|_1$$

$$\leq \max_s \mathcal{I}_{1,i}(s) \sum_{a_1 \neq a_1', a_2} \|\boldsymbol{\pi}(\boldsymbol{a}|s) - \boldsymbol{\pi}'(\boldsymbol{a}|s)\|_1$$

$$\leq \max_s \mathcal{I}_{1,i}(s) \sum_{a_1, a_2} \|\boldsymbol{\pi}(\boldsymbol{a}|s) - \boldsymbol{\pi}'(\boldsymbol{a}|s)\|_1$$

$$= \max_s \mathcal{I}_{1,i}(s) \sum_{a_1, a_2} \|[\pi_1(a_1|s) - \pi_1'(a_1|s)] \cdot \pi_2(a_2|s)\|_1$$

$$= \max_s \mathcal{I}_{1,i}(s) \sum_{a_1, a_2} \|[\pi_1(a_1|s) - \pi_1'(a_1|s)]\|_1 \cdot \pi_2(a_2|s)$$

$$= \max_s \mathcal{I}_{1,i}(s) \sum_{a_1} \|\pi_1(a_1|s) - \pi_1'(a_1|s)\|_1$$

$$= \max_s \mathcal{I}_{1,i}(s) \|\pi_1(.|s) - \pi_1'(.|s)\|_1$$

$$= \|\pi_1 - \pi_1'\|_{\mu_i}$$

$$\square$$

*Proof of Lemma 3.* For a fixed stationary policy $\pi_2$ of $\mathscr{P}_2$, the game is an MDP for $\mathscr{P}_1$. Let us define

$$v \triangleq V_i^{\pi_1, \pi_2}, \qquad v' \triangleq V_i^{\pi_1', \pi_2}, \qquad p \triangleq P_i^{\pi_1, \pi_2}, \qquad p' \triangleq P_i^{\pi_1', \pi_2},$$

so that $v, v' \in \mathbb{R}^{|\mathcal{S}|}$ are column vectors representing the unique value function for the given policy pairs, and $p, p' \in \mathbb{R}^{|\mathcal{S}| \times |\mathcal{S}|}$ are row-stochastic matrices. Although we can't directly use the results of

Even-Dar and Mansour [2003], we can apply norm inequalities to obtain:

$$\|v - v'\|_\infty \le \|v - v'\|_1 \qquad\qquad \text{(norm property)}$$
$$= \gamma \|pv - p'v'\|_1 \qquad\qquad \text{(by definition)}$$
$$= \gamma \|pv - p'v + p'v - p'v'\|_1 \qquad\qquad \text{(addition of zero)}$$
$$\le \gamma \|pv - p'v\|_1 + \gamma \|p'v - p'v'\|_1 \qquad\qquad \text{(triangle inequality)}$$
$$= \gamma \|(p - p')v\|_1 + \gamma \|p'(v - v')\|_1 \qquad\qquad \text{(linear algebra)}$$
$$\le \gamma \|p - p'\|_1 \|v\|_\infty + \gamma \|p'\|_1 \|v - v'\|_\infty \qquad\qquad \text{(Hölder inequality)}$$
$$\le \gamma \|\pi_1 - \pi_1'\|_{\mu_i} (1 - \gamma)^{-1} + \gamma \|v - v'\|_\infty,$$

In the above, we define the matrix norm $\|p\|_1 \triangleq \max_s \|p(\cdot|s)\|_1$ to be the row-wise induced matrix norm, where the last part is due to equation (A.1), the boundedness of the rewards in $[0, 1]$, and the fact that $p'$ is a row-stochastic matrix so $\|p'\|_1 = 1$. Rearranging, we obtain

$$\|v - v'\|_\infty \le \gamma \|\pi_1 - \pi_1'\|_{\mu_i} (1 - \gamma)^{-2}. \qquad\qquad (A.2)$$

To conclude, note that $u_i(\pi_1, \pi_2) = \sigma^\top v$ and $u_i(\pi_1', \pi_2) = \sigma^\top v'$. The symmetrical result is obvious. $\qquad\square$

*Proof of Lemma 4.* Let $\bar{\boldsymbol{\pi}} = (\bar\pi_1, \bar\pi_2)$ be the optimal joint policy for $\widehat{\mathcal{G}}$. Then $\bar\pi_i$ is also the optimal response to $\bar\pi_j$, as the game is fully co-operative. If $\mathscr{P}_1$ fixes $\bar\pi_1$, it selects a specific MDP for $\mathscr{P}_2$. In $\mathcal{G}$, agent $\mathscr{P}_2$ response will only be optimal according to its own model $\mu_2$. However, for any fixed Markov policy of one agent, the original ($\mathcal{G}$) and modified ($\widehat{\mathcal{G}}$) game are $\epsilon$-equivalent MDPs for the other agent, with respect to $L_1$ [Even-Dar and Mansour, 2003, Def. 2], where $\epsilon = \|\mu_1 - \mu_2\|_1$. By applying Even-Dar and Mansour [2003, Lemma 4], which states that the optimal policy in the approximate MDP induces a $2\epsilon(1 - \gamma)^{-2}$-optimal policy in the true MDP, we obtain the claim. In particular, the cited Lemma 4 states that for $\epsilon$-equivalent MDPs, the optimal policy for one MDP is $2\epsilon(1 - \gamma)^{-1} \cdot V_{\max}$-optimal for the other. In our case, $V_{\max} \le (1 - \gamma)^{-1}$ as the rewards are bounded in $[0, 1]$. Substituting $\epsilon$ gives us the result. $\qquad\square$

*Proof of Theorem 1.* We begin by noting that

$$\|\pi_1^\alpha - \hat\pi_1\|_{\mu_1} = \max_{s \in \mathcal{S}} \|\pi_1^\alpha(\cdot \mid s) - \hat\pi_1(\cdot \mid s)\|_1 \mathcal{I}_{1,1}(s) \qquad\qquad (A.3)$$

$$\|\pi_1^\alpha(\cdot \mid s) - \hat\pi_1(\cdot \mid s)\|_1 = \sum_a |\pi_1^\alpha(a \mid s) - \hat\pi_1(a \mid s)|_1 \qquad\qquad (A.4)$$

$$= \sum_a |\alpha\bar\pi_1(a \mid s) + (1 - \alpha)\hat\pi_1(a \mid s) - \hat\pi_1(a \mid s)|_1 \qquad\qquad (A.5)$$

$$= \alpha \|\bar\pi_1(\cdot \mid s) + \hat\pi_1(\cdot \mid s)\|_1. \qquad\qquad (A.6)$$

Replacing, we obtain

$$\|\pi_1^\alpha - \hat\pi_1\|_{\mu_1} = \alpha \max_{s \in \mathcal{S}} \|\bar\pi_1(\cdot \mid s) - \hat\pi_1(\cdot \mid s)\|_1 \mathcal{I}_{1,1}(s) \qquad\qquad (A.7)$$

$$= \alpha \|\bar\pi_1 - \hat\pi_1\|_{\mu_1} \qquad\qquad (A.8)$$

Combining with Lemma 3, we have $u_1(\pi_1^\alpha, \hat\pi_2) \ge u_1(\hat\pi_1, \hat\pi_2) - \alpha \|\bar\pi_1 - \hat\pi_1\|_{\mu_1} C$. Combining with the theorem's hypothesis,

$$u_1(\pi_1^\alpha, \hat\pi_2) + \alpha \|\bar\pi_1 - \hat\pi_1\|_{\mu_1} C \ge u_1(\hat\pi_1, \hat\pi_2) = u_1(\bar\pi_1, \hat\pi_2) + \Delta$$
$$u_1(\pi_1^\alpha, \hat\pi_2) \ge u_1(\bar\pi_1, \hat\pi_2) + \Delta - \alpha \|\bar\pi_1 - \hat\pi_1\|_{\mu_1} C.$$

Let us now define

$$\alpha^* \triangleq \min \left\{ \alpha \mid \pi_2^{\mathrm{B}}(\pi_1^\alpha) = \hat\pi_2 \forall \alpha \in [\alpha^*, 1] \right\}$$

to be the smallest mixing coefficient for which $\mathscr{P}_2$ sticks to $\hat\pi_2$. Then the achievable improvement over $\bar\pi_1$ is

$$\Delta - \alpha^* C \|\bar\pi_1 - \hat\pi_1\|_{\mu_1}. \qquad\qquad (A.9)$$

We can characterise $\alpha^*$ by noting that, by Lemma 3, for $\mathscr{P}_2$:

$$u_2(\pi_1^\alpha, \hat{\pi}_2) \geq u_2(\bar{\pi}_1, \hat{\pi}_2) - (1-\alpha)\left\|\bar{\pi}_1 - \hat{\pi}_1\right\|_{\mu_2} C$$

$$u_2(\pi_1^\alpha, \pi_2) \leq u_2(\bar{\pi}_1, \pi_2) + (1-\alpha)\left\|\bar{\pi}_1 - \hat{\pi}_1\right\|_{\mu_2} C.$$

$\mathscr{P}_2$ will not switch to any other deterministic $\pi_2 \neq \hat{\pi}_2$ as long as $u_2(\pi_1^\alpha, \hat{\pi}_2) > u_2(\pi_1^\alpha, \pi_2)$. For this, it is sufficient that:

$$u_2(\bar{\pi}_1, \hat{\pi}_2) - (1-\alpha)\left\|\bar{\pi}_1 - \hat{\pi}_1\right\|_{\mu_2} C \geq u_2(\bar{\pi}_1, \pi_2) + (1-\alpha)\left\|\bar{\pi}_1 - \hat{\pi}_1\right\|_{\mu_2} C$$

$$\delta \geq 2(1-\alpha)\left\|\bar{\pi}_1 - \hat{\pi}_1\right\|_{\mu_2} C.$$

As this means that $\mathscr{P}_2$ responds with $\hat{\pi}_2$ for all $\alpha \geq 1 - \delta/(2C\left\|\bar{\pi}_1 - \hat{\pi}_1\right\|_{\mu_2})$, we conclude that $\alpha^* \leq 1 - \delta/(2C\left\|\bar{\pi}_1 - \hat{\pi}_1\right\|_{\mu_2})$. Replacing in (A.9) completes the proof. $\qquad\square$

Figure 2: Counterexample. Blue arrowed-lines indicate AI model, red lines human model. Black lines indicate agreement, with dashes indicating a stochastic transition. The transitions from state 3 and 4 are identical and are represented by the subgraph to the right of the dotted line.

*Proof of Theorem 2.* This follows from a counterexample with 7 states and action sets $\mathcal{A}_1 = \{A, B\}$, $\mathcal{A}_2 = \{C, D\}$, shown in Figure 2. Notice that in every state, at most one agent's action affects the outcome.

In state 1, $C$ leads to state 4 and $D$ to state 3, but the human thinks the converse is true. In state 2, both players agree that there is a 0.5 probability of reaching either 3 or 4. These two states have identical transition probabilities. However, the AI knows that if it chooses $A$, the next state is 7, and if it chooses $B$, the next state is 6. The human disagrees, and thinks $B$ leads to the "bad" state 5.

Consequently, it is advantageous for the AI to commit to playing $B$ if the players arrive at state 4 from state 1, otherwise to commit to playing $A$ from both states 3 and 4. Thus, the optimal AI policy (as well as the value of a state) is history-dependent.

$\qquad\square$

# B  Experimental Setup

Two parameters, the discount factor and the horizon, are the same for both domains. They are set to 0.95 and 100, respectively. The other parameters are problem dependent and are described below.

**Multilane Highway.** The multilane highway problem is described by a 5 lane road. We have 4 basic types of states: $A$ (starting state), $B$ (destination state), $\{C_{ij}\}$ for $i \in \{1, \ldots, I\}$, $j \in \{1, \ldots, J\}$, where $I = 5$ is the number of lanes in the highway and $J = 5$ is the discretized length of each lane, and $D$ (accident state). Since our reward functions is only state-dependent, to model the fact that the human can intervene, we double every state, except $A$. In one of the two states in each pair, the reward takes into account that the human intervenes.

There are 4 basic types of transitions: $A \to C_{i,1}$, $C_{i,j} \to C_{i',j+1}$, $C_{i,5} \to B$, and $(A-or-C_{i,j}) \to D$, but only 5 (active) actions, which correspond to choosing $i'$ in $C_{i',j+1}$ (if $B$ is the next state, all

the actions lead to $B$). Transitions $A \to C_{i,1}$, $C_{i,j} \to C_{i',j+1}$, and $C_{i,5} \to B$, happen with a high probability; otherwise, the transitions end in $D$. Therefore, all the transition probabilities can be defined via the *accident probability* (transitioning to $D$), which in the experiments is selected as $\alpha \cdot i/(5+1)$, with $\alpha = 0.001$ — small $\alpha$ corresponds to a small accident probability. The lanes with smaller index $i$ are safer, however, we also model them to be more expensive because more time is needed to traverse them. Furthermore, the human overestimates $\alpha$ by an *error factor*, which in our experiments ranges from 0 to 50.

Before we describe how the human's interventions are modeled, we define the reward when the human does not intervene. Rewards are equal to: 0 in state $A$, 10 in state $B$, $-10$ in state $D$, $-10 \cdot \beta \cdot (1 - i/(5+1))$ in state $C_{i,j}$, where $\beta$ is a *cost factor*, which ranges from 0 to 0.2 in our experiments. This implies that safer lanes lead to smaller penalty.

To model the human's intervention, we extend the human agent with an additional passive action. The human's active actions always override the AI, so the AI can act only when human is not active. This implies that $i$ in the $C_{i,j}$ state is chosen from the human's action if the human is active, and otherwise, the $AI$'s action determines it. We also need to define the cost of intervention (the cost of the human not selecting the passive action). In this case, it is defined to be equal (in absolute value) to the penalty in $C_{i,j}$, so the overall reward is negative. Notice that the interventions have different costs in different states. For $B$ and $D$, we consider it to be equal to $C_{i,j}$ with $i = 6$.

**Food and Shelter.** We are in a $5 \times 5$ grid world, with the shelter located at a fixed location and food randomly appearing at one of the five top locations. Whenever we pick food, it reappears in one of the remaining locations at the top. The reward for food is equal to 1, the penalty for the shelter being destroyed is $-0.1$, and the starting point is the location of the shelter. As shown in Figure 1(d), boundaries of the world are surrounded by the wall as well as some parts of the world near food locations. The experimental setup follows the food and shelter domain from Guo et al. [2013], with the following differences:

- Our world size is $5 \times 5$.
- We allow diagonal moves.
- Error in movement happens with probability equal to 0.1 and takes the agent uniformly at random to one of 8 neighbouring locations.
- We introduce a human with the ability to intervene by overriding the AI's actions, with cost of intervention $\beta$ that we vary from 0 to 0.5 in the experiments.
- The human has an incorrect error model— in particular, the transition probabilities (of a move) are skewed toward uniformly random move by a factor:
  - $\alpha$ for non-diagonal moves (i.e. the probability of the correct move is modulated by $1 - \alpha$);
  - $2\alpha$ for diagonal moves (i.e. the probability of the correct move is modulated by $1 - 2\alpha$);

  where $\alpha$ is a parameter that we vary between 0 and 0.5 in the experiments.

## C  Algorithm Pseudocode

Pseudocode for the backwards induction algorithm sketched in the main text is given in Algorithm 1. For the deterministic version of the algorithm, we can simply replace the linear program with a maximisation over $\mathscr{P}_1$'s actions. As this algorithm runs for $T$ steps, and it does not necessarily converge to a stationary policy (see Section 3.1), the output may be a time-dependent policy. We can then extract the best stationary policy by considering the policy $\pi(a_{t,1} \mid s_t)$ at each step $t$ of the first player.

---

**Algorithm 1:** Backwards induction for MVDPs

---
**Data:** $T$, $\mu_1$, $\mu_2$;

1 **begin**
2      $V_1 = [\rho(0), ..., \rho(S)]$;
3      $V_2 = [\rho(0), ..., \rho(S)]$;
4      $Q_1^* = [0, ..., S]$;
5      $Q_2^* = [0, ..., S]$;
6      **for** $t = T$ *to* $t = 1$ **do**
7          **for** $s \in \mathcal{S}$ **do**
8              **for** $(a_1, a_2) \in \mathcal{A}_1 \times \mathcal{A}_2$ **do**
9                  $Q_1(s, a_1, a_2) \leftarrow \rho(s) + \gamma \cdot \sum_{s'} \mu_1(s' \mid a_1, a_2, s) \cdot V_1(s')$;
10                  $Q_2(s, a_1, a_2) \leftarrow \rho(s) + \gamma \cdot \sum_{s'} \mu_2(s' \mid a_1, a_2, s) \cdot V_2(s')$;
11              $Q_1^*(s) \leftarrow -\infty$;
12              **for** $a_2 \in \mathcal{A}_2$ **do**
13                  Find policy $\pi_{Q_s}$ for state $s$ with value $Q_s$ using the LP:

$$\max_{\pi_1(.|s)} \sum_{a_1} \pi_1(a_1|s) \cdot Q_1(s, a_1, a_2)$$

$$\text{s.t. } \forall \hat{a}_2 : \sum_{a_1} \pi_1(a_1|s) \cdot Q_2(s, a_1, a_2) \geq \sum_{a_1} \pi(a_1|s) \cdot Q_2(s, a_1, \hat{a}_2),$$

$$\forall \hat{a}_1 : 0 \leq \pi_1(a_1|s) \leq 1, \text{ and } \sum_{a_1} \pi_1(a_1|s) = 1.$$

                 **if** $Q_s \neq NULL$ *and* $Q_s > Q_1^*(s)$ **then**
14                      $Q_1^*(s) \leftarrow Q_s$;
15                      $\pi_1^*(a_t \mid s_t = s) \leftarrow \pi_{Q_s}$;
16                      $\pi_2^*(a_t \mid s_t = s) \leftarrow a_2$;
17                  **else if** $Q_s \neq NULL$ *and* $Q_s = Q_1^*(s)$ **then**
18                      we randomly break the tie

19      $V_1 \leftarrow Q_1^*$;
20      $V_2 \leftarrow Q_2^*$;
21      **return** $(\pi_1^*, \pi_2^*)$

---