[Reviews · NeurIPS 2017]

Reviewer 1



This article presents a Markov decision model with two independent controllers, who have different beliefs about the transition probabilities of the model. The model is designed to fit the situation of an autonomous vehicle with both a machine and a human controller. As such the presentation is very much focused on a situation in which one player (the machine) has a correct model of state transitions whereas the other player (the human) has an incorrect model. The example of interest is where the human can, at any time, decide to intervene and seize control the game. The model allows us to consider the implications of this dichotomy, and in particular whether the "correct" controller should use a different strategy if it knows about the failures of the "incorrect" controller. Efforts are also made to calculate how much lower the system payoffs will be under this scenario. I like the concept of the paper, and think it's quite well put together. However the paper is insufficiently discursive for us to appreciate the meaning of the concepts provided. We are told about the influence of an agent, about policy distance, and sub optimality levels, but I gleaned no intuition for how tight these concepts are, or how to use them in an example setting. Furthermore the model results in non-Markovian and complicated solution concepts, and the presented solution method is only approximate with no guarantees on its level of sub optimality. Some experiments are carried out, but they are cursory and it's nigh on impossible to work out what is going on. In summary, each component is (to my mind) interesting, but the presentation of all of these concepts in such a tight package means that we do not learn enough about any of them to ascertain if they are important. Furthermore, a discrete state and action space is so distant from the application area that it is difficult to see that this is actually sufficiently along the lines towards the application to merit giving up on complete theory.

Reviewer 2



The paper introduces multi-view decision processes (MVDP) as a cooperative game between two agents, where one agent may have an imperfect model of the transition probabilities of the other agent. Generally, the work is motivated well, the problem is interesting and clearly relevant to AI and the NIPS community. The theoretical results are nicely complemented by experiments. Presentation is mostly smooth: the main goals are consistent with the narrative throughout the paper, but the some details from the experiments seem less clear (seem points below.) Some points are confusing and need further clarification: 1. In the second footnote on page 3, what is a "worst-case best response"? This is not defined, seems like they should all share the property that they attain the maximum value against some policy. 2. The authors talk about solving using backward induction and make reference to Bozansky et al 2016 and Littman 1994, but those works treat exclsively the zero-sum case. MVDP is a cooperative game that can be at least as hard as general-sum games (by Lemma 1), and backward induction generally cannot solve these due to equilibrium selection problems leading to non-unique values for subgames, also shown in Zinkevich et al 2005. How is it possible to apply backward induction by propagating single Q and V values here then? The authors claim to find an approximate joint optimal policy, which is stronger than a Nash equilibrium, and it seems by Zinkevich et al 2005 this should at least require cyclic (nonstationary) strategies. 3. From the start the authors talked about stochastic policies, but this is a cooperative identical payoff game. It's not clear later (seeing the proof of Lemma 1) that mixed strategies are necessary due to the different views leading to an effectively general-sum game. How then could a the determinstic version of the algorithm that computes a pure stationary policy be approximately optimal? Even in the Stackelberg case, mixing could be necessary. Minor: The role of c in the c-inervention games is not clear. On page 6, the authors write a cost c(s) is subtracted from the reward r_t when P2 takes an action other. But then in Remark 1 gives the equation rho(s, a) = [rho(s) + cI{a_{t,2} != a^0}] / (1+c) does not subtract c. This can be easily fixed by rewording the text at bottom of page 6.

Reviewer 3



Summary: Motivated by settings in which a human and an artificial intelligence agent coordinate to complete a task (e.g in autonomous driving of a vehicle), authors consider a Stackleberg game in which the leader (AI) has perfect knowledge of the transition probabilities, but the follower’s (human) model (his/her subjective probabilities) might be inaccurate. Authors analyze the impact of the inaccuracy in the follower’s subjective transition probabilities on policy planning by the leader and present a dynamic programming approximation algorithm for computing the optimal policy.
 I find the problem interesting, relevant, and well-motivated. But I find it hard to interpret the results authors present. In particular, section 2 of the paper quantifies how far the optimal utility can be from that of the jointly optimal policy and the naive optimal policy, but these bounds are not elaborated on and is hard to understand whether they say anything non-trivial and/or interesting. The algorithm proposed (which is a form of backward induction) does not solve the planning problem exactly (that is shown to be computationally hard as one may expect), but it additionally suffers from multiple limitation (as outlined by authors themselves in paragraph “optimality”, line 235).That calls for substantial expansion of the empirical section of the paper, possibly evaluating on real-world data and settings.